# Cyclic Peptides in Pipeline: What Future for These Great Molecules?

**DOI:** 10.3390/ph16070996

**Published:** 2023-07-12

**Authors:** Lia Costa, Emília Sousa, Carla Fernandes

**Affiliations:** 1Laboratório de Química Orgânica e Farmacêutica, Departamento de Ciências Químicas, Faculdade de Farmácia, Universidade do Porto, Rua de Jorge Viterbo Ferreira, 228, 4050-313 Porto, Portugal; liapdacosta@gmail.com; 2Interdisciplinary Centre of Marine and Environmental Research (CIIMAR), Edifício do Terminal de Cruzeiros do Porto de Leixões, Av. General Norton de Matos s/n, 4050-208 Matosinhos, Portugal

**Keywords:** bioactivity, clinical trials, cyclic peptides, cyclization, pipeline

## Abstract

Cyclic peptides are molecules that are already used as drugs in therapies approved for various pharmacological activities, for example, as antibiotics, antifungals, anticancer, and immunosuppressants. Interest in these molecules has been growing due to the improved pharmacokinetic and pharmacodynamic properties of the cyclic structure over linear peptides and by the evolution of chemical synthesis, computational, and in vitro methods. To date, 53 cyclic peptides have been approved by different regulatory authorities, and many others are in clinical trials for a wide diversity of conditions. In this review, the potential of cyclic peptides is presented, and general aspects of their synthesis and development are discussed. Furthermore, an overview of already approved cyclic peptides is also given, and the cyclic peptides in clinical trials are summarized.

## 1. Introduction

Peptides, molecules that contain two or more residues of amino acids linked by an amide bond [1], can be considered to fall between small molecules and large biological molecules, such as proteins or antibodies [2]. Peptides have several advantages over small molecules used in conventional therapy, such as high selectivity, potency, biotarget specificity, few side effects, and low accumulation in tissues [3,4]. When compared to proteins and antibodies, peptides have the advantage of a lower immunogenicity [2].

Over the years, the therapeutic potential of peptides has been exploited for a broad spectrum of biological activities, such as antimicrobial, antihypertensive, antioxidant, anticancer, antidiabetic, and anti-inflammatory, among others, which attract the attention of the pharmaceutical [5,6,7], cosmeceutical [8], and nutraceutical [9,10] industries.

The era of therapeutic peptides began with the first medical administration of insulin in the 1920s [11]. This discovery revolutionized the treatment of patients with type I diabetes, and peptides were seen as potential therapeutic tools [12]. About 40 years later, the first hormones used in clinical practice, oxytocin, and vasopressin, were synthesized [13]. Some industrial groups have dived into this field, and the interest in this type of molecules increased [14]. While the advantages of using these molecules were explored, their limitations also emerged at a time when the discovery and development of small molecules was at its peak. This circumstance has led to the stagnation of research on peptides as drugs. Despite this, peptides continued to be explored as tools for the study of targets, and in the 1980s, the interest in these molecules returned backed by biotechnology companies [14]. Since then, the tendency of approved peptides as therapeutic agents has been increasing [5].

In addition to their therapeutic potential, peptides can also be used in imaging and disease diagnosis [5]. For example, it is known that peptides composed of arginine-glycine-aspartate (RGD) moiety have an affinity to bind to integrins, which are heterodimeric receptors that play pivotal roles in cells. By binding to integrin, RGD peptides can prevent angiogenesis, a process involved in diseases, such as cancer and rheumatoid arthritis [15,16,17]. These peptides can also be used for tumor imaging when linked to radioisotopes or even to create tumor-targeted drug delivery systems reducing the adverse effects inherent to conventional chemotherapy [18,19].

Until May 2023, one hundred and fourteen peptides have been approved by the regulatory authorities as therapeutic agents (Figure 1), which included pharmaceuticals and diagnostic tools [20].

Among the approved peptides, the cyclic peptides represent 46% of the total approvals (Figure 2A) [21]. Gramicidin S (antibiotic) was the first cyclic peptide to be used as a drug. Its discovery in 1944 by Gause and Brazhnikova during the Second World War and its use in Soviet military hospitals revolutionized the field of cyclic peptides [22,23]. Other interesting examples are telavancin, dalbavancin, and oritavancin (semi-synthetic cyclic lipoglycopeptide antibiotics) [24], anidulafungin (from the class of echinocandin antifungals) [25], lanreotide, pasireotide, and romidepsin (anticancer drugs), and linaclotide (derived from an enterotoxin for gastrointestinal (GI) disorders) [26]. The last approved cyclic peptide was rezafungin (antifungal analog of anidulafungin), being approved by Food and Drug Administration (FDA) in 2023 [27]. This drug is administered orally and has a half-life of 30 h, which represents an advance in pharmacokinetic characteristics in comparison with other peptide drugs. In fact, regarding the routes of administration of approved peptides (Figure 2B), parenteral administration is the most frequent, of which the intravenous route is the most recurrent [21].

Another relevant application of peptides is in drug delivery, considering the good hydrophilicity/hydrophobicity ratio, as well as intra and intermolecular interaction of amino acids by weak non-covalent bonds, which makes them capable of organizing themselves to form nanostructures. Peptide nanostructures have demonstrated a great drug load capacity and drug protection and are responsive to external stimuli [28]. Recently, new cyclic peptides exhibiting nanospherical structures demonstrated the ability to form stable complexes with short-interfering RNA (siRNA), proving to be a promising tool in nucleic acid delivery for cancer treatment, as an example [29].

Moreover, cyclic peptides, such as vancomycin, teicoplanin, and ristocetin (macrocyclic antibiotics), in addition to their therapeutic actions, were also explored as chiral stationary phases for chromatographic applications [30,31]. This application of macrocyclic antibiotics in liquid chromatography was introduced by Armstrong et al. in 1994 [32]. The high number of stereogenic centers and the macrocyclic structure of these peptides allow a variety of interactions with the analytes to enantioseparate and the possibility of forming inclusion complexes, which contribute to their high capacity of chiral recognition [33,34].

Cyclic peptides can be obtained from natural sources, both terrestrial and marine [35,36]. With regard to terrestrial sources, these can be from animal origin [37], such as venoms [38] (a rich source of bioactive peptides revised in [39]), plants [40], microorganisms [41], among others). Examples of peptides obtained from terrestrial sources are the antibiotics vancomycin (isolated from the soil bacterium *Amycolatopsis orientalis*) [42], daptomycin (from the soil bacterium *Streptomyces roseoporus*) [43], teixobactin (from the soil bacterium *Eleftheria terrae*) [44], and apamin (isolated from bee *Apis mellifera*) [45].

Bioactive marine cyclic peptides can be found in marine tunicates [46,47], sponges [47,48], algae [49], bacteria [50,51], cyanobacteria [52,53], fungi [54,55], and other invertebrates [56], including symbionts [57] and non-symbiotic microorganisms, such as sponge-associated fungi [58,59]. Marine-derived peptides display a broad spectrum of bioactivities [60,61], mainly anticancer [62] and antimicrobial [63] being one of the research topics that gives a very high output, with a considerable increase in the number of publications (268 per year), from 2010 to 2020 [64]. To highlight that, one marine cyclic peptide-derived drug has reached the market—ziconotide [65] (the first FDA-approved marine peptide, in 2004).

Chemical synthesis is also a remarkable source of peptides, which allows obtaining an appropriate amount of compound to carry out further large-scale biological assays, including studies of the mechanism of action, pharmacokinetics, toxicity, and others [66,67,68,69]. In the literature, several reports can be found describing synthetic routes for peptides, including very large and complex structures [70,71]. In addition, the synthesis allows for obtaining the structurally diverse analogs and derivatives of the natural peptides with improved properties for structure–activity relationship (SAR) studies [72]. The synthetic strategies for molecular modifications can include the following: (1) incorporation of non-proteinogenic amino acids to prevent proteolysis of peptides [73]; (2) acetylation of the *N*-terminus of short peptides to increase peptidase stability in serum and, consequently, enhance the half-life [74]; (3) glycosylation to improve protein–protein interaction, protein permeability, metabolic stability, and bioavailability [75]; lipidation to enable the binding to a carrier serum protein and, consequently, enhance the half-life, among others [76,77].

Typically, there are two strategies to synthesize peptides: solution-phase and solid-phase peptide synthesis (SPPS) [63]. They include two key steps in the formation of a peptide bond between two amino acids: the activation of the carboxyl group by coupling agents and the use of temporary protecting groups to direct the reaction to the desired direction [78]. The two most used strategies are fluorenylmethyloxycarbonyl (Fmoc)/*tert*-butyl (tBu) and *tert*-butyloxycarbonyl (Boc)/benzyl (Bn) strategies [79]. In addition to classical peptide synthesis techniques [69], significant efforts have been carried out for the introduction of sustainable and innovative processes for synthesis and purification methodologies [80,81].

Despite several advantages of peptides, they also have characteristics that are disadvantageous when they are used for drug development, mainly concerning pharmacokinetic issues: (1) they have a short half-life in the plasma because of the action of peptidases; (2) are easily degraded by enzyme actions and pH hydrolysis on the GI tract, which makes them not bioavailable orally; (3) have low membrane permeability, which makes its passage through membranes on absorption locals and intracellular biotargets difficult [2]. Nevertheless, the pharmacokinetic properties of these molecules can be improved with some strategies of molecular modifications, such as conjugation with polyethylene glycol (PEG), albumin, or proteins, as well as other approaches, such as cyclization [14].

Cyclization of peptides has proven to be an asset in enhancing the advantages of linear peptides, but also a way to overcome their disadvantages. Cyclization of a peptide reduces the spatial vibrations of the molecule leading to a decrease in conformational changes. In addition, cyclization induces an increase in the surface area available for interaction with the biological target. These two reasons lead to an increase in binding affinity and selectivity to the target [4]. In addition to pharmacodynamic considerations, the pharmacokinetic properties of peptides can also be improved by cyclization, as the rigidification of the structure leads to a lowering of the energy barrier required for the peptide to adapt to the membrane environment and bind to transport proteins to enter the cell by passive diffusion or active transport. Thus, cyclization can improve the absorption and membrane permeability of peptides [5]. Cyclization of peptides also gives greater metabolic stability, as cyclic peptides are resistant to the action of exopeptidases, due to the lack of terminal amine and carboxylic acid groups, and endopeptidases, by blocking the access to the cleavage site [19]. Although cyclization may improve the pharmacokinetic properties of peptides, it is important to highlight that for many cyclic peptides, the poor pharmacokinetic parameters are one of the main reasons for failure in phase I/II trials [82]. Through chemical synthesis, it is possible to carry out molecular modifications to obtain derivatives with improved characteristics [14].

Despite these concerns, the unique characteristics of cyclic peptides make them attractive molecules to be explored. Proof of this finding are the diverse examples of approved cyclic peptides widely used in therapeutics for different diseases.

## 2. Cyclization Strategies

Cyclization can occur in four different ways, depending on the type of constituent amino acids: head-to-tail cyclization, head-to-side chain cyclization, side-chain-to-tail cyclization, and side-chain-to-side-chain cyclization [83].

The cyclization step is one of the most challenging reactions in peptide synthesis [84]. Examples of the most relevant peptide cyclization methods are summarized in Figure 3.

Cyclization is conventionally performed in solution using coupling agents (Figure 3A) but has several drawbacks, such as the possibility of C-terminal epimerization, oligomerization, and low coupling efficiency according to the ring size and peptide sequence. In addition, the fact that the linear peptide must be fully protected to proceed with cyclization makes it difficult to be soluble in organic solvents [85,86]. To improve coupling efficiency, these reactions are usually performed at high dilution to prevent oligomerization. To avoid C-terminal epimerization, additives can be used, or the strategy of using glycine (Gly) or proline (Pro) residues as activation sites can be adopted [72]. Nevertheless, these strategies are not fully effective, and other approaches may be necessary to increase the efficiency of cyclization, allowing a chemoselective bond between two amino acid residues. Furthermore, the probability of C-terminal epimerization occurring is reduced, and the full protection of the side chains is not required [85].

Native chemical ligation (NCL) is a technique based on a chemoselective ligation between a cysteine (Cys) residue of *N*-terminal and another thioester peptide by a nucleophilic attack that forms an intermediate with a thioester ligation. This intermediate undergoes an intramolecular reaction and rearranges itself to form a native peptide bond (Figure 3B) [87]. In 2012, Zheng et al. [88] reported the chemical synthesis of cyclotide kalata (KB1) using this approach.

NCL-desulfurization is a variation of the NCL method that consists of removing the thiol group (desulfurization) after the reaction and replacing it with alanine (Ala) or other amino acid residues. This is performed as most naturally occurring cyclic peptides do not have Cys residues on their structures [72]. Yan [89] used this strategy for the synthesis of the antibiotic peptide Microcin J25.

Serine (Ser)/Threonine (Thr) ligation approach consists of a chemoselective ligation between a peptide with C-terminal glycolaldehyde ester and another peptide with Ser, Thr, or Cys on *N*-terminal [90]. Although this method was an evolution in chemoselective bond formation, the reaction was very slow when the *N*-terminal end contained Thr or Ser, and, in addition, conversion of the formed ring into an amide bond did not occur [91]. Due to this difficulty, another version of Ser/Thr ligation was proposed using salicylaldehyde ester on C-terminal instead of the glycolaldehyde ester (Figure 3C) [91]. The usefulness of the Ser/Thr ligation for peptide cyclization has been demonstrated in the synthesis of daptomycin [92], cyclomontanin B [93], yunnanin C [94], and mahafacyclin B [95].

KAHA ligation approach consists of a formation of an amide bond by a powerful and chemoselective ligation reaction between *N*-alkylhydroxylamines and *C*-terminal peptide α–keto carboxylic acids. These two groups react chemoselectively to form native amide bonds that do not require reagents and only produce carbon dioxide and water (Figure 3D) [96]. The peptide antibiotic gramicidin was synthesized via this method [97].

Staudinger’s ligation approach consists of a reaction of C-terminal phosphinothioester with *N*-terminal azide to form an iminophosphorane that is hydrolyzed to form an amide bond (Figure 3E) [98,99].

Another synthetic possibility is the on-resin cyclization by previous anchoring of the linear peptide to a resin (Figure 3F) [100]. This method takes advantage of the effect known as ‘‘pseudo-dilution’’ with intramolecular reactions being favored over intermolecular reactions, thus allowing a reduction in the occurrence of epimerization and oligomerization. In addition, the by-products formed are easily removed by filtration and washing [86]. The cyclic peptide stylostatin 1 [101] and teixobactin analogs [102] were successfully synthesized using this strategy.

Enzyme-mediated cyclization (Figure 3G) is another strategy considering that enzymes are highly chemoselective, which is an asset for cyclization reactions without the formation of side products, with the ability to catalyze a cyclization reaction with a low amount of enzyme relative to substrate [103]. In addition, they are non-toxic and have a low cost [85]. Butelase 1 [104], sortase A [105], subtiligase [106], and transglutaminase [107] are examples of enzymes used for peptide cyclization. For instance, the cyclization of bacteriocins AS-48 and uberolysin was catalyzed by butelase 1 [108].

Recently, new developments in cyclization methods have been described [109]. As examples, we highlight the CyClick strategy, which consists of the formation of a cyclic imine formed between the *N*-terminus of the peptide and the *C*-terminus of an aldehyde group (Figure 3H) [110,111]. This reaction places the amide in the second position near the imine, which leads to a nucleophilic attack by the amide nitrogen on the imine forming the 4-imidazolidinone linked to the cyclized peptide. No by-products are found, so the yields reported for this approach were high and were found to be chemoselective [111].

In 2022, Nwajiobi et al. [112] reported a new strategy that consists of incorporating *para*-aminophenylalanine into a linear peptide by the SPPS method, following the addition of HCl and sodium nitrite, and then drastically changing the pH to 7.5 to obtain a diazonium ion arene (Figure 3I). Secondary amines, such as Pro or monomethyllysine, react with this ion to form a triazene bond. The yield is quantitative, and the reaction takes only 5 min and does not give rise to by-products. This is a chemoselective reaction for secondary amines, which takes place in a single step and is reagent-free [112].

Another efficient strategy was developed by Vanjari et al. [113] and consists of cyclizing propargylated peptides through imine formation mediated by gold (Figure 3J). This method allows the cyclization of peptides without the need for protecting groups and in a very short time (only 30 to 60 min). The gold complex reacts with the triple bond of the propargyl group. A free amine of the peptide then acts as a nucleophile by attacking the gold-activated alkyne. This reaction occurs by a Markovnikov addition mechanism that is regioselective [113].

Chemoselective ligations and these more recent cyclization methods may be powerful alternative tools for the synthesis of cyclic peptides considering their advantages over cyclization in solution. Nevertheless, it is important to infer that these synthetic strategies also have limitations, such as the need of the presence of specific functional groups in the amino acid residues for cyclization to occur.

## 3. Strategies for Discovering and/or Optimizing Cyclic Peptides

As previously mentioned, cyclic peptides have more advantages over linear peptides, and there are several natural cyclic peptides in therapeutics without structural modifications. Despite this, cyclic peptides may have problems related to their pharmacological properties, mainly due to pharmacodynamic or pharmacokinetic properties. To improve the characteristics of natural cyclic peptides, diverse strategies can be adopted, involving, for example, amino acid substitutions and modifications on the peptide skeleton [114].

Some natural amino acid residues are more vulnerable to the action of enzymes, such as aspartate (Asp), asparagine (Asn), and methionine (Met) [115]. By replacing these amino acid residues with others less susceptible, it is possible to obtain a cyclic peptide more resistant to enzymatic action and, therefore, with a longer half-life [116]. In addition, it is also possible to improve the binding affinity of the peptide to the biotarget [117].

Another strategy may be the replacement of natural L-amino acids with D-amino acids that are not naturally present in proteins and, thus, will not be recognized by proteolytic enzymes [118]. In addition, an improvement in the half-life can be achieved, with the cyclic peptide hormone somatostatin being a representative example. This hormone has a half-life of only three minutes in vivo, which limits its clinical application. Therefore, analogs were synthesized, aiming to maintain biological activity but with a longer half-life. These analogs were developed by molecular simplification and amino acids substitution [119]. It was found that octreotide, an analog comprising a lower number of amino acids and containing a D-configuration tryptophan (Trp) residue, showed a 200-fold increase in potency and a 30-fold increase in half-life in relation to somatostatin [119]. This peptide is used in clinical practice for the treatment of acromegaly [120].

*N*-Methylation of the backbone is a modification that improves the oral bioavailability of peptides [121]. For example, cyclosporine A has *N*-methylations on its backbone and is one of the orally bioavailable cyclic peptides in the therapeutics [122]. Eric Biron and his group [123], inspired by cyclosporine A, developed *N*-methylated analogs of somatostatin, which they believed to be sufficient, together with cyclization, for developing orally active analogs of this hormone. Nevertheless, from thirty-one analogs, only seven have shown good affinity to the target. These seven analogs were used for bloodstream uptake and enzyme stability testing in rats, as well as for permeability studies in the Caco-2 model. Only a tri-methylated peptide showed a high apparent permeability [123].

Another strategy that can be adopted is the replacement of *α*-amino acids with *β*-amino acids, which allows an improvement in enzyme degradation stability, membrane permeability, and bioactivity [124]. Gademann et al. [125] have synthesized a cyclo-β tetrapeptide analog of somatostatin that has demonstrated the ability to mimic the natural peptide displaying biological activity on the nanomolar order for human receptors.

Moreover, the introduction of semicarbazide moieties in the peptidic skeleton, giving rise to azapeptides or modifications in the carbohydrate portions of glycopeptides are strategies used to enhance the pharmacokinetic properties and activity of cyclic peptides [126,127]. Modifications in the carbohydrate moiety of vancomycin gave rise to an analog with antibacterial activity against vancomycin-resistant species [127].

Although the trend of discovery of new cyclic peptides in recent years has been mainly focused on the modification of natural compounds, with the evolution of technology, computers were put at the service of discovering new molecules with different biological activities [128]. Computer-aided methods constitute rational planning of new drugs, making it possible to perform virtual screening of a large number of molecules [129]. For example, by the de novo design method, Frecer et al. [130] reported synthetic cationic peptides with potent antibacterial activity, and Hosseinzadeh et al. [131] described cyclic peptides with high binding affinity to the surface of proteins.

Nowadays, display techniques are also an important tool for the discovery of new cyclic peptide drugs. There is a great diversity of display techniques, both in vivo and in vitro, capable of generating and identifying peptides, as well as testing the interaction between proteins or peptides and ligands. Display systems include phage, yeast, mRNA, ribosome, bacterial, and DNA display [14]. Nowadays, this approach has an important role in peptide drug discovery because it allows the creation of a non-natural peptide library with improved pharmacokinetic properties and lower toxicity [132].

Phage display was reported for the first time in 1985 by George P. Smith [133]. In general, this approach is used to probe protein–ligand interactions based on binding affinity, where the protein is “displayed” on the surface of a phage particle [134]. Phage display is a selection method widely used for drug screening, principally in peptide-screening [135,136]. It is both an in vivo and in vitro method because peptides are displayed in vivo, but affinity tests are performed in vitro [134]. The first step is to construct a phage library with lots of phage clones with different peptides displayed on the surface. This is performed by manipulating the phage DNA to express DNA of random peptide sequences of a peptide library on the coat protein gene of phage particles. To do this, many phages can be used, and the coat protein chosen to express the peptide is usually protein pIII. After this process, the desired peptide is expressed in the coat protein [137]. Then, the phage display libraries can be tested against one molecule, the immobilized target, to test the affinity of the binding [138]. The chosen phages are eluted and amplified by infecting more bacteria to achieve more quantity of phages with the interesting peptides to make this cycle more times. The cycle is repeated three–five times, and the peptide that binds to the target is identified by DNA sequencing (Figure 4) [136,138].

This approach allows testing peptide libraries with millions of different peptides to find interactions between new peptides and the targets and isolate and identify the peptides of interest [138]. Phage display is a technique, which is easy to use, has a good relationship between cost and quality, and allows a fast and high throughput screening of new peptides [134,138]. One disadvantage of this method is that *Escherichia coli* (bacteria used in this approach) is not able to express some eukaryotic proteins. As a consequence, the diversity of peptides obtained by phage display is around 10^9^ [136].

The display technique can also be used to generate and identify cyclic peptides [14]. The first library of cyclic peptides obtained by phage display was formed by peptides that contained a Cys group at each end, which reacted by forming a disulfide bridge after oxidation [139]. Peginesatide is a cyclic peptide discovered by phage display and was approved to treat secondary anemia induced by chronic kidney disease [140]. This molecule is a dimer of a peptide discovered by phage display linked with a linker to two PEG, which allowed it to increase its half-life in vivo. However, it was withdrawn from the market after one year of its approval due to anaphylactic reactions [140,141].

It is also possible to obtain bicyclic peptides through phage display, and one approach is to use a library of linear peptides modified with three Cys groups. After these peptides are expressed in the coat protein, they react with 1,3,5-tris(bromomethyl)benzene (TBMB), forming a bicyclic peptide [142].

## 4. Cyclic Peptides in the Pipeline

Currently, there are various cyclic peptides in clinical development to cover such diverse medical conditions as cancer, infectious diseases, and hematological disorders, among others. Table 1 summarizes 27 cyclic peptides in clinical trials, including their generic names, therapeutic indication(s), current stage of development, name of the pharmaceutical company responsible for the development, and source (natural or synthetic).

The 21 new cyclic peptides that are being used in clinical trials for the first time for a particular disease or condition are described here. The reported structures of these molecules are illustrated in Figure 5, and their development and clinical aspects are detailed in the next sections.

### 4.1. POL6326—Balixafortide

Balixafortide (POL6326) is a cyclic synthetic peptide with 14 amino acids developed by Spexis. It is a very potent and selective antagonist of chemokine receptor C-X-C chemokine receptor type 4 (CXCR4), inhibiting tumor growth and metastasis, as well as activating the immune response in the tumor microenvironment [178,179]. Balixafortide was developed by using a protein epitope mimetic (PEM) technology created by Polyphor (now Spexis) to discover and optimize fully synthetic CXCR4 inhibitor cyclic peptides [180]. The development process of cyclic peptides followed a rational approach since the structures mimic the natural β-hairpin loop of the target protein combined with phage display as a tool to find new hits for further modification and optimization [181].

POL6326 has a half-life of 6 to 8 h and demonstrated high potency and selectivity in preclinical trials [182]. In December 2018, a phase I clinical trial began to assess the efficacy, safety, and tolerability of balixafortide administered with eribulin versus eribulin alone in the treatment of human epidermal growth factor receptor type 2 (HER2) negative, locally recurrent or metastatic breast cancer (ClinicalTrials.gov identifier: NCT01837095) [182,183]. In the same month, a phase III clinical trial began (ClinicalTrials.gov identifier: NCT03786094) [184]. Nevertheless, in August 2021, the end of this study was announced because it was found that the administration of balixafortide together with eribulin does not significantly increase the activity of eribulin alone; therefore, the initial hypothesis of this study failed [185]. Recently, in July 2022, Spexis revealed that balixafortide demonstrated to be more effective when given together with docetaxel, when compared with their administration alone, in a preclinical model of prostate cancer [186]. Additionally, the potential of this cyclic peptide is also being explored for applications in solid and hematological tumors, as well as in rare non-oncological diseases [186]. Furthermore, balixafortide has already shown promising results in vitro, exhibiting cell protection in a SARS-CoV-2-induced cytopathic effect assay with a half maximal effective concentration (EC_50_) of 10 µM. These promising results have also been confirmed in vivo, where the amount of virus in hamster blood after SARS-CoV-2 infection was significantly reduced [143]. These preliminary results suggested that it may constitute a new treatment for COVID-19.

### 4.2. BL-8040—Motixafortide

Motixafortide (BL-8040) is a selective inhibitor of the CXCR4 receptor developed by BiolineRx. This peptide has a high binding affinity (IC_50_ ≈ 1 nM) and long receptor occupancy when compared to other inhibitors of this receptor [187,188].

Motixafortide is a synthetic cyclic peptide with 14 amino acids whose development began with molecular modifications of a natural defense peptide, polyphemusin. Applying this strategy, three inhibitory analogs of the CXCR4 receptor with activity against human immunodeficiency virus (HIV) were obtained [189,190]. Among all the synthesized analogs, T140 showed the highest inhibitory activity against the entry of HIV-1 and the highest binding affinity to the CXCR4 receptor [191]. T140 peptide was synthesized by SPPS using Fmoc as a protecting group followed by the one-pot technique where trimethylsilyl chloride—dimethyl sulfoxide/trifluoroacetic acid (TFA) were used simultaneously to cleave the peptide from the resin, deprotect the protecting groups and form the disulfide bond [191]. Since the T140 peptide had low serum stability, some analogs were synthesized, including motixafortide [189].

This peptide is currently being developed for two therapeutic applications: mobilization of hematopoietic stem cells (HSCs) for autologous transplantations and treatment of solid tumors [144]. For the treatment of solid tumors, balixafortide acts in three different ways: firstly, it increases the number of immune cells such as B and T cells and natural killer cells (on which the CXCR4 receptor is expressed) by mobilizing these cells from the bone marrow to the peripheral circulatory system, increasing the effectiveness of immunotherapy [190]; secondly, it antagonizes the CXCR4 receptor on immunosuppressive cells, thereby enhancing the natural immunosuppression of a tumor environment and modulating the tumor microenvironment [144]. Furthermore, it causes the infiltration of effector cells in the tumor microenvironment. All these results make motixafortide a promising immune-modulatory agent with potent antitumor effects [192]. This peptide completed a phase II clinical trial whose first part consisted of studying the mechanism of motixafortide administered in combination with pembrolizumab in patients with metastatic pancreatic adenocarcinoma, and the second part consisted of studying the safety, tolerability, and efficacy of motixafortide when administered in conjunction with pembrolizumab and chemotherapy. The results were promising, and the next step will be to conduct another trial to confirm the data before moving on to phase III clinical trials [190]. In 2020, another phase-II clinical trial was initiated to evaluate the administration of motixafortide with cemiplimab and standard-of-care chemotherapy in patients with metastatic pancreatic ductal adenocarcinoma [144].

Regarding the mobilization of HSCs, motixafortide has completed phase-III clinical trials and is in the pre-submission phase. It is expected to obtain FDA approval and begin commercialization of motixafortide as a stem cell mobilization agent for bone marrow transplantation in multiple myeloma patients as early as 2023 [144].

### 4.3. BT1718

BT1718 is a bicycle drug conjugate with antitumor activity developed by Bicycle Therapeutics. This conjugate is composed of a 13–amino acids bicyclic peptide covalently linked to a toxin (DM1), a potent anti-tubulin agent [146,147]. The peptide part of this conjugate binds with high affinity and specificity to membrane type I matrix metalloproteinase (MT1-MMP) [147]. MT1-MMP is a metalloprotease that is overexpressed in several solid tumors and is directly related to tumor invasion and metastasis [145]. This metalloprotease is used as a target to facilitate the delivery of DM1 [193]. BT1718 was obtained from libraries of compounds generated by phage display. These peptide sequences were then cyclized with a TBMB scaffold, and those with the highest affinity for MT1 were identified through a high-throughput selection process. These peptides were further optimized with the introduction of non-natural amino acids to improve the plasma stability of the peptide [145].

BT1718 is currently in phase I/IIa clinical trial in which the maximum safe dose, the potential side effects, and pharmacokinetic properties in patients with advanced solid tumors are being studied (ClinicalTrials.gov Identifier: NCT03486730) [146].

### 4.4. BT8009

BT8009 is a bicyclic drug conjugate developed by Bicycle Therapeutics composed of a 14–amino acids bicyclic peptide targeting Nectin-4 linked to monomethyl auristatin E (MMAE), a cytotoxin by a valine–citrulline linker [149]. Nectin-4 is a molecule that is overexpressed in certain types of tumors but limited in healthy tissues [148]. The peptide binds to Nectin-4 on the cell surface, and the cytotoxin is released via a peptidase. The binding between the peptide and Nectin-4 revealed a high affinity [148]. The peptide was identified using phage display technology and was further synthesized to increase affinity and improve its hydrophilicity and stability [148]. A phase I/II clinical trial is currently underway, in which its administration in combination with nivolumab is being studied to assess safety, efficacy, and possible adverse effects (NCT04561362) [149].

### 4.5. BT5528

BT5528 is another bicyclic peptide conjugated to MMAE via a cleavable linker with antitumor activity developed by Bicycle Therapeutics. In this way, the 14-amino acids bicyclic peptide binds to EphA2 and delivers MMAE mainly in cancer cells [150]. Similar to peptides of the two previous conjugates developed by Bicycle Therapeutics, this bicyclic peptide was obtained by phage display screen against EphA2. Cyclization of the phage display generated a peptide sequence that was further cyclized using 1,3,5-triacryoyl-1,3,5-triazinane, a more hydrophilic scaffold than TBMB followed by lead optimization to improve stability and increase hydrophilicity using non-natural and polar charged amino acids [194]. Its synthesis was performed on Rink amide resin using Fmoc solid-phase synthesis guided by X-ray crystallography studies [194]. This peptide is now in phase I/II clinical trial to evaluate the safety and tolerability of BT5528 alone and in conjunction with nivolumab in patients with advanced solid tumors associated with EphA2 expression (NCT04180371) [195].

### 4.6. VT1021

VT1021 is a cyclic pentapeptide developed by Vigeo Pharmaceuticals with antitumor activity. This peptide is derived from prosaposin (a glycoprotein) that stimulates the production of thrombospondin-1 (Tsp-1) in myeloid-derived suppressor cells (MDSCs). Tsp-1 binds to CD36 of M2 macrophages, transforming them into M1 macrophages with anticancer action [196]. Tsp-1 also binds to tumor cells that express the CD36 receptor, inducing their death by apoptosis. This peptide activates cytotoxic T lymphocytes and inhibits angiogenesis [197,198]. In this way, the tumor microenvironment is reprogrammed, becoming an inhibitory environment for the tumor. Thus, this mechanism of action may be a successful approach for the treatment of tumors that have not responded to conventional chemotherapy [199].

In preclinical trials, this peptide demonstrated strong antitumor activity in animal models of ovarian, pancreatic, and breast cancer and exhibited the ability to cause tumor regression and reprogram the tumor microenvironment [197]. VT1021 has completed the phase I/II clinical trial safety, pharmacology, and preliminary efficacy in patients with advanced solid tumors (ovarian, pancreatic, triple-negative breast cancer, glioblastoma, and tumors that overexpress CD36) that did not respond to conventional therapies (ClinicalTrials.gov Identifier: NCT03364400). VT1021 was demonstrated to be safe and well-tolerated and showed activity in reprogramming the tumor microenvironment in patients expressing the CD36 receptor [151,196]. Due to these positive results, Vigeo announced in January of 2022 that VT1021 will advance to a phase II/III clinical trial in patients with glioblastoma that is already underway (ClinicalTrials.gov Identifier: NCT03970447). Furthermore, it is planned to proceed with efficacy studies in additional solid tumor indications [200].

### 4.7. ALRN-6924

ALRN-6924 is a synthetic cell-penetrating stapled α-helical peptide with 18 amino acids that mimics the transactivation domain of α-helix of p53, thus binding with high affinity to murine double minute 4 protein (MDMX) and murine double minute 2 protein (MDM2) (inhibitory constant (K_i_) values of 24.7 and 7.7 nM, respectively) [201], thus inhibiting MDMX/p53 and MDM2/p53 protein–protein interactions [202,203].

The ALRN-6924 peptide is an analog of the ATSP-7041 [204] peptide that was the lead compound in a series of compounds emerging from phage display peptide optimization. The ATSP-7041 analog was synthesized using the Fmoc SPPS strategy. The linear peptide was synthesized using HCTU/6-Cl-1-hydroxybenzotriazole (HOBt). Then, the olefin building block was bounded and introduced with HATU/HOAt. The Fmoc groups were removed with 20% (*v*/*v*) piperidine/DMF, and the *N*-terminal amine was acetylated. Finally, cyclization was performed with the linear peptide still bound to the resin. Lastly, the resin was removed, and the final product was purified by liquid chromatography-mass spectrometry (LC-MS) [205].

This peptide has been developed by Aileron Therapeutics as a chemoprotective agent for healthy cells to minimize the adverse effects caused by chemotherapy [206]. This is possible by taking advantage of the fact that the TP53 gene that encodes the p53 protein is mutated in 50% of cancers [202]. That is, ALRN-6924 binds to MDMX and MDM2 and leaves p53 free, thus temporarily activating cell-cycle arrest. This binding only occurs in cells in which p53 is not mutated, in healthy cells. Therefore, tumor cells continue to be active and remain susceptible to the action of antitumor agents [207]. In this way, this peptide makes it possible to improve the effectiveness of conventional chemotherapeutic agents, protecting healthy cells and thus improving the usual side effects [152]. In tumors where the p53 mutation is rare, ALRN-6924 has also been shown to be useful since releasing p53 from the MDMX and MDM2 proteins induces cell cycle arrest and subsequent apoptosis of these tumor cells [208].

Currently, ALRN-6924 is in phase Ib clinical trials in which its safety, tolerability, and chemoprotective effect are being studied in combination with paclitaxel in patients with advanced, metastatic, or unresectable solid tumors (ClinicalTrials.gov identifier: NCT03725436) [209].

### 4.8. CEND-1

CEND-1 is an eight-amino acid cyclic peptide developed by CEND Therapeutics. This peptide was identified by phage display from a library of cyclic peptides whose cyclization occurred between two Cys residues and was shown to recognize tumor blood vessels in experimental metastasis mouse models of human prostate cancer [210]. The strategy used aimed to increase the tissue penetration of a tumor-targeted antitumor drug and decrease the resistance to antitumor drugs created by the high density of tissue around the tumor [153]. This is possible thanks to the presence of an RGD motif that selectively binds to the β5 integrin that is present exclusively in the tumor blood vessels and not in the blood vessels of healthy tissues [211]. Once bound to integrin, CEND-1 undergoes the action of a proteolytic enzyme that gives rise to the CendR portion, which increases the binding affinity to neuropillin-1, activating a transcytotic pathway in endothelial cells that increases the increased vascular transcytosis and permeability. For these reasons, this peptide allows the possibility to be used as an antitumor drug to reach tumor cells more easily [153].

Currently, a phase II clinical trial is underway, which is intended to study the effect of CEND-1 when administered with chemotherapy in patients with untreated metastatic pancreatic ductal adenocarcinoma (ClinicalTrials.gov Identifier: NCT05042128) [212].

### 4.9. POL7080—Inhaled Murepavadin

Murepavadin (POL 7080) is a fully synthetic cyclic peptide with 14 amino acid residues developed by Spexis [213]. This peptide has a unique antimicrobial activity being the first member of a new class of antibiotics with novel mechanisms of action—Outer Membrane Protein Targeting Antibiotics (OMPTA)—developed by Polyphor in collaboration with the University of Zurich [214]. Murepavadin has a nonlytic mechanism of action, acting through binding to an outer membrane protein responsible for the biosynthesis of lipopolysaccharide in Gram-negative bacteria—lypopolysaccharide transport protein D (LptD). Through the link established with LptD, murepavadin inhibits the transport of lipopolysaccharide, causing changes in the bacterial outer membrane, which leads to cell death [213,215]. Murepavadin has a specific activity for *Pseudomonas aeruginosa* being explored for infections caused by this bacterium in patients with cystic fibrosis [213].

The development of this peptide was inspired by the host defense peptides, which have a broad spectrum of antimicrobial action and low resistance against Gram-negative bacteria—protegrin-I (a natural peptide with a *β*-sheet structure due to disulfide bridges) [216]. As these peptides had unfavorable properties to be used as drugs, the strategy was to synthesize a library of compounds inspired by protegrin-I using the PEM approach developed by Polyphor [217]. Compound libraries with improved plasma stability and drug-like properties were synthesized, where murepavadin emerged [218]. Synthesis of the analogs was carried out using SPPS to synthesize the linear precursor in a 2-chlorotrityl chloride resin using Fmoc chemistry. The cyclization was carried out in solution using *N*-[(dimethylamino)-1 *H*-1, 2,3-triazolo [4,5-b]pyridin-1-ylmethylene]-*N*-methylmethanaminium hexafluorophosphate *N*-oxide (HATU)/1-hydroxy-7-azabenzotriazole (HOAt) as coupling agent after cleavage of the resin. Purification was carried out by preparative HPLC to obtain the desired cyclic peptide [217,219].

In in vitro and in vivo preclinical trials, its activity against various *Pseudomonas* species and other Gram-negative bacteria, the potential development of resistance, pharmacodynamics, pharmacokinetics, and metabolism were studied [213,215]. The pre-clinical results were promising, which made this peptide advance to various phase-I and II clinical trials where its safety, efficacy, tolerability, pharmacokinetics, and pharmacodynamics, among other aspects, were tested [213,220]. However, the phase III trial was temporarily stopped due to safety concerns (ClinicalTrials.gov Identifier: NCT03409679). Considering these unexpected results, Spexis initiated, in 2021, phase-I clinical trials maintaining the same goal, the treatment of *P. aeruginosa* infections in patients with cystic fibrosis, but testing inhaled murepavadin [214]. *P. aeruginosa* is characterized by a biofilm-forming mode of growth, resistance to antibiotics, and a high capacity to mutate. This mode of administration may be a good strategy for the treatment of infections caused by this microorganism. One of the reasons is that it acts immediately in the lungs without having to undergo systemic exposure, which also allows adverse effects to be minimized [154].

### 4.10. Thanatin Derivatives

In 2019, Spexis, together with the University of Zurich, started a program to develop the thanatin derivatives [214]. Thanatin is an insect-derived 21 amino acid residue peptide that shows activity against Gram-positive and Gram-negative bacteria, as well as against some fungi [221]. This antibiotic belongs to the new OMPTA class, but unlike murepavadin, it binds to lypopolysaccharide transport protein A (LptA), inhibiting lipopolysaccharide transport and causing damage to the microorganism’s outer membrane [222].

Thanatin has been undergoing optimization to improve its low in vitro plasma stability and its low half-life time in vivo. The successive optimizations have resulted in several thanatin derivatives with improved drug-like properties. This process is being developed by Spexis with funding from CARB-X [214].

### 4.11. CD-101—Rezafungin

Rezafungin (CD-101) is a new semisynthetic echinocandin that was under development for the treatment of fungal infections of *Candida* and *Aspergillus* spp. [223]. Rezafungin is an anidulafungin analog, which is considered a second-generation echinocandin that presents improved pharmacokinetic and pharmacodynamic parameters. It also has an improved tissue penetration and a good safety profile [224,225]. The mechanism of action of rezafungin is similar to other echinocandins; it is an inhibitor of cell wall synthesis by inhibition of 1,3-β-D-glucan synthesis [224]. Rezafungin completed a phase-III clinical trial that proved to be safe and effective in the treatment of candidemia and/or invasive candidiasis (ClinicalTrials.gov Identifier: NCT03667690) and is currently participating in a phase-III clinical trial to assess the efficacy and safety in the prevention of invasive fungal diseases (ClinicalTrials.gov Identifier: NCT04368559). Recently, rezafungin for injection was approved by FDA to treat candidemia and invasive candidiasis in adults with limited or no alternative treatment options [27].

### 4.12. RA-101495—Zilucoplan

Zilucoplan (RA-101495) is a 15-amino acid macrocyclic synthetic peptide developed by Ra Pharmaceuticals (now UCB Pharma). It is an inhibitor of the C5 complement that binds allosterically with a subnanomolar affinity [157,226]. This peptide is under development for the treatment of paroxysmal nocturnal hemoglobinuria and generalized myasthenia gravis (gMG) [157]. Zilucoplan plays an important role in these two diseases since it binds with high affinity and specificity to complement component 5 (C5), preventing cleavage in C5a and C5b and the interaction between C5b and C6. In this way, the membrane attack complex (MAC) is not formed, and the cell membrane is not destroyed [226].

Zilucoplan was synthesized by SPPS Fmoc/tBu method. The synthesis was performed on a Liberty automated microwave peptide synthesizer. The coupling of the amino acids was performed using 2-(6-chloro-1-*H*-benzotriazole-lyl)-l,l,3,3,-tetramethylaminium hexafluorophosphate (HCTU) as a coupling agent and diisopropylethylamine (DIEA) as a base. The cyclization of zilucoplan was via a lactam bridge between two amino acids of the linear peptide [227].

Zilucoplan completed two phase-II clinical trials for therapeutic application in paroxysmal nocturnal hemoglobinuria that served to study the safety and efficacy of this drug in patients with this disease and in patients who had an inadequate response to eculizumab (ClinicalTrials.gov Identifier: NCT03030183). Another phase-II clinical trial started with this drug to evaluate the effect of its prolonged use in patients who completed the previous clinical trials; however, UCB Pharma decided to stop the investigation of zilucoplan for application in the paroxysmal nocturnal hemoglobinuria disease [228]. For the gMG application, zilucoplan completed a phase-II clinical trial (ClinicalTrials.gov identifier: NCT03315130) in 2020, which demonstrated effective complement inhibition and a favorable safety and tolerability profile [229]. This peptide has already successfully completed a phase-III clinical trial (ClinicalTrials.gov identifier: NCT04115293), in which its efficacy, safety, and tolerability in patients with gMG were confirmed [230]. Considering these positive results, UCB Pharma plans to move forward with regulatory submissions for this drug during the current year [231]. A phase-III clinical trial (ClinicalTrials.gov identifier: NCT04225871) is currently ongoing, which is an open-label extension study to assess the long-term safety, efficacy, and tolerability of zilucoplan in patients with gMG who have already completed a clinical trial with zilucoplan [228].

### 4.13. AP301—Solnatide

Solnatide (AP301) is a fully synthetic cyclic peptide with 17 amino acids developed by Apeptico [158,232]. This peptide is under development for the treatment of pulmonary permeability edema in patients with moderate-to-severe acute respiratory distress syndrome (ARDS). Regarding the route of administration, it is intended to be administered by inhalation of a liquid aerosol delivered directly to the lungs with minimal plasma absorption [233]. Its structure mimics the lectin-like domain of human tumor necrosis factor and has the ability to activate the amiloride-sensitive sodium channel (ENaC), which is the main driving force capable of absorbing water through the reabsorption of Na+ through the lung epithelium, thus promoting the alveolar liquid clearance [234]. In addition, solnatide is also able to reduce intracellular levels of reactive oxygen species, inhibit the activation of protein kinase C **α**, reduce the degree of phosphorylation of the myosin light chain, and inhibit the pro-inflammatory pathway by reducing the production of cytokines [233].

In a non-clinical trial phase, solnatide was demonstrated to promote alveolar liquid clearance through the activation of ENaC in dog, pig, and rat models, reduced the long injury in a pig model, and proved to be effective in primary graft dysfunction after transplantation in rats [235]. In a phase-I clinical trial, it exhibited a safe and well-tolerated profile in increasing single doses [236]. In a phase-IIa clinical trial, solnatide was demonstrated to be safe in patients with mild to severe ARDS and appeared to be more effective in sicker patients, this evidence will be confirmed in later clinical trials [237].

Although this peptide is still in phase II clinical trials, it has already been approved for use in patients with COVID-19 in Austria and Italy in June 2022. The reason was due to the urgent need for drugs to reduce mortality and the severity of the disease caused by the SARS-CoV-2 virus. Solnatide was tested in patients infected with this virus because its infection causes pulmonary permeability edema. Interestingly, it demonstrated an improvement in symptoms and an increased survival rate after 28 days [233].

### 4.14. POL6014—Lonodelestat

Lonodelestat (POL6014) is a synthetic 13-amino-acid cyclic peptide developed by Polyphor through PEM technology [159]. Development and commercialization rights for this peptide were obtained by Santhera Pharmaceuticals from Polyphor. POL6014 was synthesized by SPPS and is in development for cystic fibrosis, acting as a potent and selective inhibitor of neutrophil elastase. By inhibiting this enzyme, the number of neutrophils released is reduced, thus decreasing inflammation and lung tissue damage, which preserves lung function [159]. This peptide is administered by inhalation [238]. Lonodelestat ended phase Ib/IIa clinical trial in December 2020 (ClinicalTrials.gov Identifier: NCT03748199).

### 4.15. THR-149

THR-149 is a bicyclic peptide developed by Oxurion NV that prevents pathologies associated with diabetic macular edema (DME). This peptide is an inhibitor of plasma kallikrein (PKal) that leads to the production of mediators of inflammation, vasodilation, and increased vessel permeability leading to microvascular retinal damage, among other effects [161]. THR-149 emerged from a library of PKaI inhibitor compounds obtained by phage display technology combined with a chemical cyclization [160]. THR-149 was the most potent and stable [161]. The synthesis process of this peptide starts with the synthesis of the linear peptide by Fmoc SPPS using Symphony automated peptide synthesizer and Rink amide resin. Then, the peptide was cleaved from the resin, precipitated, and lyophilized. The final step involved cyclization with TBMB [160].

Currently, this peptide is in a phase IIb clinical trial where its safety and efficacy are being evaluated (ClinicalTrials.gov Identifier: NCT04527107).

### 4.16. PTG-300—Rusfertide

Rusfertide (PTG-300) is a synthetic 18-amino acids hepcidin mimetic peptide developed by Protagonist Therapeutics [239]. This peptide acts on ferroportin, an iron transporter, causing its internalization and degradation [239]. This effect causes a decrease in the level of iron in the serum and transferrin saturation because it prevents iron from leaving the cells responsible for its storage [162,239]. This peptide is under development for the treatment of symptoms of polycythemia vera, a rare hematologic disease in which erythropoiesis occurs at very high levels, which increases the risk of thrombosis and causes iron deficiency [162,240]. Due to its proven activity in iron regulation, PTG-300 is also being studied for the treatment of hereditary hemochromatosis, a disease characterized by excessive absorption of iron by cells that leads to the degradation of organs, such as the liver, heart, and pancreas [163].

Currently, a phase-III clinical trial is ongoing to evaluate the safety and efficacy of PTG-300 in the improvement of polycythemia vera symptoms (NCT05210790) [241].

### 4.17. PN-943

PN-943 is a cyclic peptide α4β7 integrin antagonist developed by Protagonist Therapeutics for the treatment of ulcerative colitis [164,165]. α4β7 is present on the surface of memory T and B lymphocytes, associated with the recruitment of leukocytes to the intestinal mucosa [165]. This peptide prevents the binding of α4β7 integrin to the mucosal addressin cell adhesion molecule that is normally expressed in high values in inflamed GI tissues, which leads to the impediment of leukocyte recruitment and, consequently, to the reduction in GI inflammation [165].

PN-943 was found to be stable in the GI tract after oral administration, which was attractive because its site of action is in the GI tract [165]. This drug is in phase-II clinical trials to evaluate the safety, tolerability, and clinical efficacy of two different doses in patients with moderate to severe active ulcerative colitis (ClinicalTrials.gov Identifier: NCT04504383).

### 4.18. PL8177

PL8177 is a synthetic cyclic seven-amino-acid peptide that acts as a potent and selective agonist of the melanocortin 1 receptor developed by Palatin Technologies [166]. It is an analog of α-melanocyte-stimulating hormone, which is a peptide that regulates central and peripheral inflammation. Thus, this peptide is under development for the treatment of ulcerative colitis by oral administration [166,242]. This peptide contains a D-phenylalanine (Phe), a strategy used to improve the pharmacokinetic properties of peptides [166].

A phase-II clinical trial is underway to study its safety, tolerability, efficacy, pharmacokinetics, and biomarkers of oral administration in patients with ulcerative colitis (ClinicalTrials.gov Identifier: NCT05466890).

### 4.19. PA-001

PA-001 is a constrained cyclic peptide initially discovered by PeptiDream, but which has been optimized and selected by PeptiAID [167]. This peptide showed very strong activity against the original SARS-CoV-2 and its α, β, γ, and δ mutations. Furthermore, this peptide demonstrated synergistic effects when administered in combination with other drugs approved for emergency use for COVID-19 [167]. In pre-clinical trials, it has shown great safety [167]. In February 2022, this peptide began initial clinical trials in Japan, where it demonstrated favorable safety and pharmacokinetic results. In addition, it was also shown to be effective for the omicron variant of the SARS-CoV-2 virus (Japan Registry of Clinical Trials Trial ID: jRCTs031210601) [243]. Currently, PeptiAID intends to start a phase-I clinical trial in the United States of America and a phase-I/IIa clinical trial in Japan [243]. PeptiDream uses the Peptide Discovery Platform System, where libraries of compounds are rapidly generated by translating a DNA library into peptides and then tests the affinity of those peptides against the intended target [244].

### 4.20. AZP-3813

AZP-3813 is a macrocyclic peptide composed of 16 amino acids that acts as a growth hormone receptor antagonist. It is under development for the treatment of acromegaly, which is caused by excess secretion of growth hormone, which stimulates excess production of insulin-like growth factor 1 (IGF1). This peptide is derived from peptide sequences discovered using a cell-free in vitro transcription–translation system that was then optimized by rational design [168,245]. In preclinical trials, this peptide has been shown to be effective in decreasing IGF1 levels, which shows its potential to develop a drug against acromegaly [245].

### 4.21. PL9643

PL9643 is a synthetic cyclic peptide which is a melanocortin receptor agonist developed for the treatment of dry eye disease [169]. This peptide is in phase III clinical trials to evaluate the safety and efficacy of its ophthalmic administration to patients with dry eye disease (ClinicalTrials.gov Identifier: NCT05201170).

In addition to the peptides as new chemical entities, there are also cases of repositioning of cyclic peptides, that is, approved cyclic peptides that are in clinical trials for the treatment of other diseases. The structures of these peptides are shown in Figure 6.

### 4.22. PM90001—Plitidepsin

Plitidepsin (PM90001) is a marine-derived depsipeptide that was approved in Australia for the treatment of multiple myeloma [170]. The Plitidepsin discovery process started with its isolation from a Mediterranean marine tunicate (*Aplidium albicans*) and later was fully synthesized [170]. PharmaMar has developed this peptide, which interacts with eukaryotic Elongation Factor 1A2 (eEF1A2), thus showing antitumor action by cell cycle arrest, apoptosis, and growth inhibition [246]. In addition to its approved activity, it has also demonstrated activity against the SARS-CoV-2 virus. This peptide inhibits the host eEF1A2 used by the SARS-CoV-2 virus to replicate. Thus, plitidepsin has demonstrated in vitro an inhibitory effect on the replication of this virus [46].

Currently, plitidepsin is in phase-II clinical trials to evaluate its efficacy in immunocompromised patients with symptomatic COVID-19 (ClinicalTrials.gov Identifier: NCT05705167).

### 4.23. APL-2—Pegcetacoplan

Pegcetacoplan (APL-2) is composed of two cyclic peptides linked together by a linear chain of polyethylene glycol molecules that increases its half-life [247]. It is a C3 inhibitor developed by Appelis and was approved for the treatment of paroxysmal nocturnal hemoglobinuria in May 2021 [171]. Clinical trials are currently underway with pegcetacoplan in phases II and III of development for diseases such as amyotrophic lateral sclerosis (ALS), immune-complex membranoproliferative glomerulonephritis and C3 glomerulopathy (IC-MPGN and C3G), cold agglutinin disease (CAD) and hematopoietic stem cell transplant-associated thrombotic microangiopathy (HSCT-TMA) [248].

There are cases of cyclic peptides whose development has been interrupted (Figure 7) for various reasons. Some examples are mentioned in this sub-chapter.

### 4.24. Friulimicin B

Friulimicin B is a natural cyclic lipopeptide with 11 amino acids produced by the actinomycete *Actinoplanes friuliensis* with antibacterial activity against a broad range of Gram-positive bacteria, including antibiotic-resistant pathogens [172]. Its mechanism of action is based on blocking cell wall biosynthesis. Structurally, this peptide consists of a macrocyclic decapeptide with a lipidic tail [173]. This peptide did not successfully complete the phase I clinical trial due to its poor pharmacokinetic properties [249].

### 4.25. ShK-186—Dalazatide

Dalazatide (Shk-186) is a synthetic 37-amino-acid cyclic peptide that inhibits the voltage-gated Kv1.3 potassium channel and is developed to treat plaque psoriasis [175]. This peptide is a synthetic analog of ShK (a natural peptide toxin isolated from the venom of the sea anemone *Stichodactyla helianthus*) [250,251]. Dalazatide was synthesized using the Fmoc-SPPS strategy on an amide resin. Couplings were performed with 6-Cl-*N*-HOBt in the presence of DIC. The removal of the Fmoc group was carried out with piperidine in DMF containing HOBt. Cleavage of the peptide from the resin was achieved with TFA. The disulfide bridge formation was achieved by air oxidation or through the addition of a glutathione exchange system [252]. This molecule successfully completed the phase Ib clinical trial in 2015 (ClinicalTrials.gov Identifier: NCT02435342), but it was not entered in any further clinical trials [175]. However, Kv 1.3 therapeutics claims to have the peptide ready to start a phase II clinical trial for the treatment of inclusion body myositis [253].

### 4.26. Ularitide

Ularitide, a peptide with 32 amino acids, is the synthetic form of urodilatin, a human natriuretic peptide synthesized in the distal kidney tubules that has been isolated from human urine [176,254]. Cardiorentis is working on the development of this peptide for the treatment of acute decompensated heart failure, a disease characterized by the sudden onset of symptoms of heart failure being responsible for many deaths and morbidity [177,255].

In phase-I clinical trials, ularitide had shown positive results in vasodilation and renal protection [256]. In phase-II clinical trials, its safety and efficacy were studied, which demonstrated that ularitide decreases cardiac filling pressures, improves dysponea, and protects short-term renal function. Furthermore, ularitide demonstrated a good safety profile [257]. In the phase-III clinical trial, ularitide decreased cardiac-wall stress; however, it did not reduce myocardial mortality and did not influence disease progression [258].

### 4.27. CB-183,315—Surotomycin

Surotomycin (CB-183,315) is a 13-amino acid cyclic lipopeptide analog of daptomycin with a different tail. Similar to daptomycin, surotomycin shows antibacterial activity against *Clostridium difficile* through disruption of the cell membrane [259]. This peptide completed phase-III clinical trials; however, it did not advance in the pipeline because the results showed that surotomycin did not demonstrate superiority over vancomycin in terms of clinical response over time [174].

## 5. Final Remarks

Over the last few years, cyclic peptides have been growing and gaining their space as therapeutic agents performing various activities and even in other applications. This is due to their structural and metabolic stability properties, high binding affinity to targets, and low or inexistent toxicity.

Cyclic peptides come mainly from natural sources, but developments in the chemical synthesis of peptides have brought many advantages, including the possibility of large-scale production for various assays and the possibility of introducing molecular modifications for the synthesis of analogs and derivatives with improved pharmacokinetic and/or pharmacodynamic properties.

Computational and in vitro methods, such as phage display, allow the creation of more compounds with great structural diversity, thus increasing the chemical research space. These methods allow for the development of libraries of cyclic peptides that can be the target of a screening from which new hits can come for the development of new drugs for different applications.

The cyclization process of these molecules is a critical step of the synthesis and has some limitations, such as C-terminal epimerization, dimerization, or low cyclization efficiency. Although cyclization methods via chemoselective bonds already exist, they require the presence of specific chemical groups. Enzymatic cyclization has also been explored and is an area of great interest due to all the advantages brought by enzymes. More recently, new methods of cyclization of peptides have emerged that try to overcome all the difficulties encountered previously; even so, there is still a need for a search of methods that facilitate this step.

All the evolution in this area is reflected by the number of cyclic peptides that are in development for different conditions. In this review, 27 cyclic peptides that are (or have been) in different stages of the pipeline have been discussed. Many of the peptides mentioned here have origin or inspiration from natural peptides. For example, the discovery of plitidepsin constitutes a classic example of the isolation of a bioactive molecule. Nevertheless, most of them have been fully synthesized or replaced by synthetic analogs to improve their properties. Regarding the discovery and optimization methods, most include rational approaches, such as the phage display technique, used for both identification and optimization.

Looking carefully at all the mentioned cyclic peptides, the relevance of the chemical synthesis becomes evident, particularly SPPS, as well as the great potential of rational discovery and optimization methods in the development of new cyclic peptides. Concerning pharmacological activities, the cyclic peptides under development show a wide range of activities, but it is worth highlighting a large number of peptides with antitumor or with chemoprotective or chemotherapy-enhancing activities. Activities, such as antimicrobial, including antiviral, antifungal, or antibacterial, are also areas in which cyclic peptides are being developed. Due to the emergency in obtaining drugs to combat the SARS-CoV-2 virus imposed by the global pandemic, there have been cases of drug repositioning. This scenario also occurred for cyclic peptides, with positive cases of drug repositioning for the treatment of SARS-CoV-2 infection. The approach taken by Bicycle Therapeutics to link a cyclic peptide to a toxin, forming a conjugate with increased selectivity for the target, should also be highlighted, as this strategy seems to be a promising path to follow in the future.

## 6. Conclusions

Despite the challenges and difficulties in the development of cyclic peptides, this review demonstrates the advantages and potential of these great molecules. There is still much to explore regarding this family of compounds as new challenges are always emerging, as the case of SARS-CoV-2 infection that resulted in a pandemic that promoted intense research of numerous compounds, including cyclic peptides. In the future, it is expected that the use of cyclic peptides as drugs will increase substantially in line with technological and scientific advances.

## Figures and Tables

**Figure 1 pharmaceuticals-16-00996-f001:**
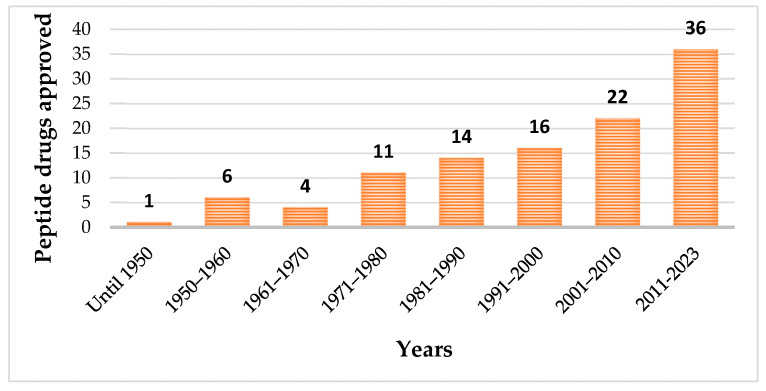
Evolution of approved peptide drugs over the years.

**Figure 2 pharmaceuticals-16-00996-f002:**
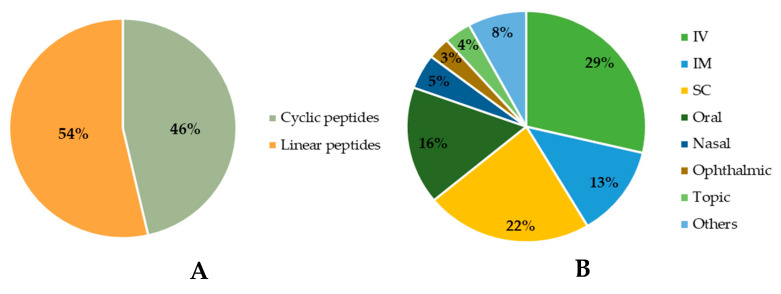
Comparison between approved cyclic and linear peptides (**A**). Routes of administration of approved peptides (**B**). IV: Intravenous; IM: Intramuscular; SC: Subcutaneous.

**Figure 3 pharmaceuticals-16-00996-f003:**
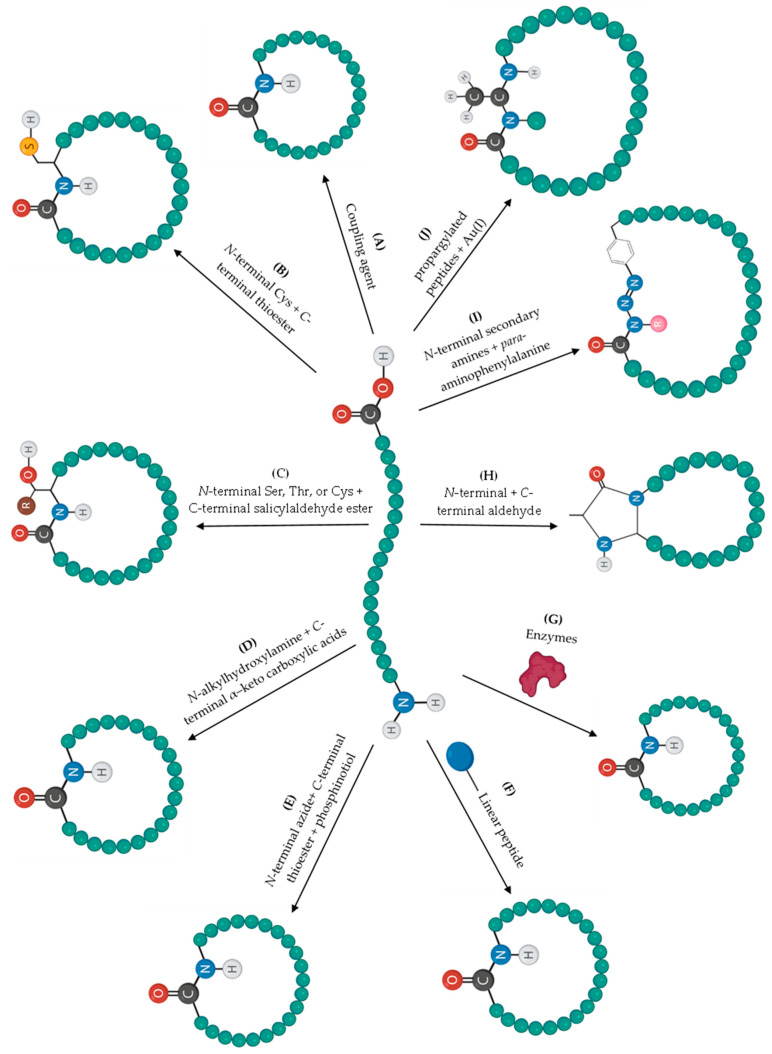
Diverse peptide cyclization methods. (**A**) Cyclization in solution; (**B**) Cyclization by native chemical ligation; (**C**) Cyclization by Ser/Thr ligation approach; (**D**) Cyclization by KAHA ligation approach; (**E**) Cyclization by Staudinger’s ligation approach; (**F**) On-resin cyclization; (**G**) Enzyme-mediated cyclization; (**H**) CyClick cyclization; (**I**) Cyclization by formation of a triazene bond; (**J**) Cyclization of propargylated peptides through imine formation mediated by gold.

**Figure 4 pharmaceuticals-16-00996-f004:**
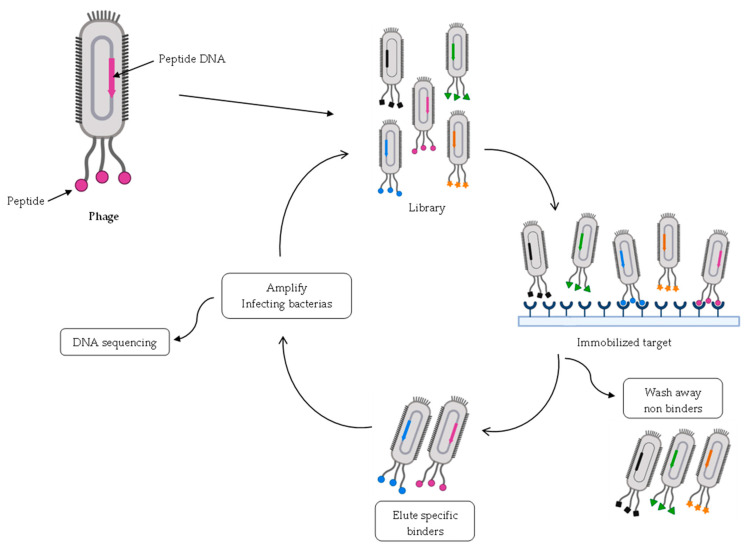
Schematic representation of the phage display strategy.

**Figure 5 pharmaceuticals-16-00996-f005:**
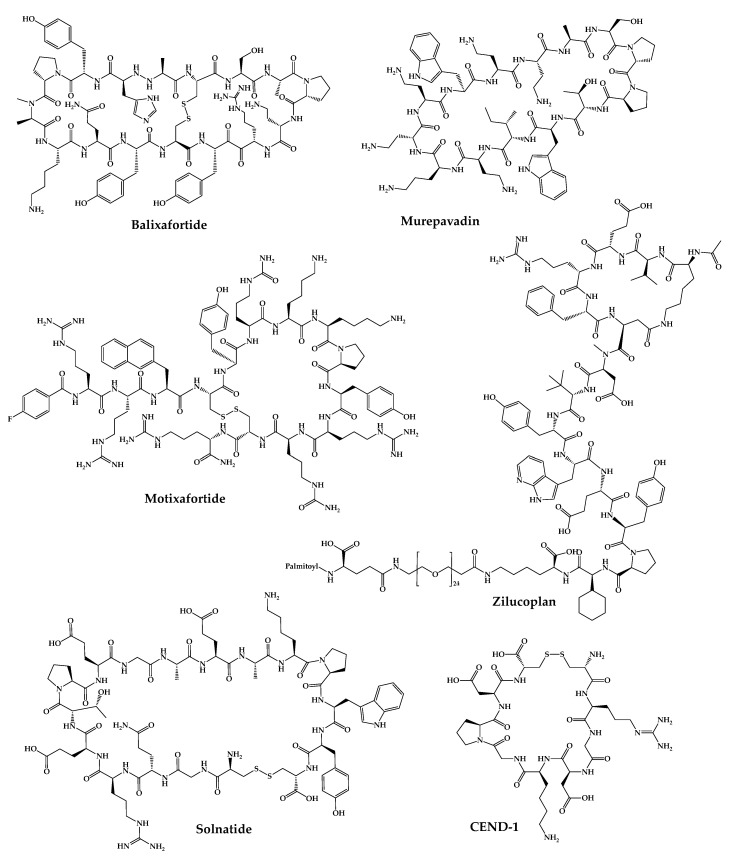
Available structures of the cyclic peptides currently in clinical trials as new chemical entities.

**Figure 6 pharmaceuticals-16-00996-f006:**
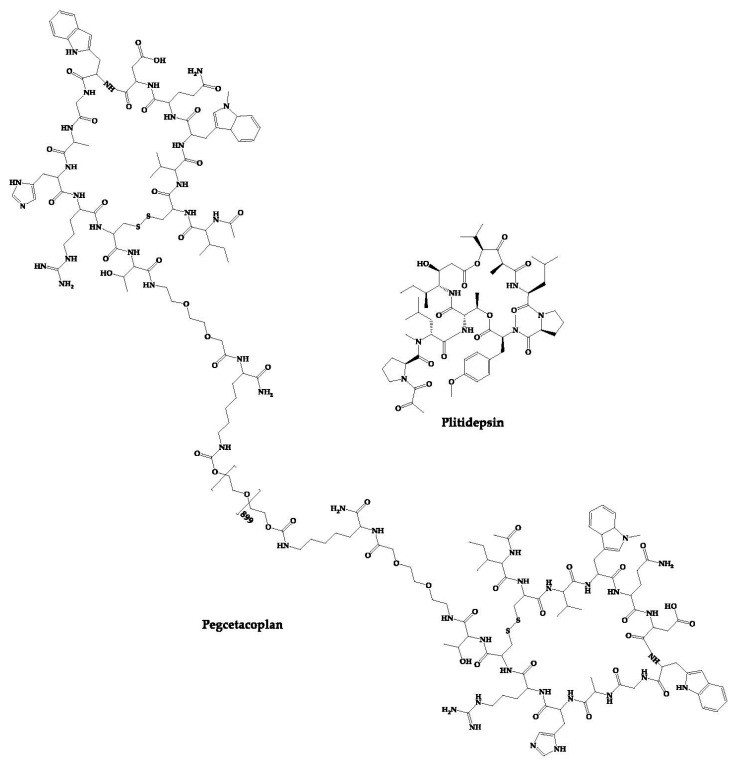
Structures of approved cyclic peptides currently in clinical trials for other diseases.

**Figure 7 pharmaceuticals-16-00996-f007:**
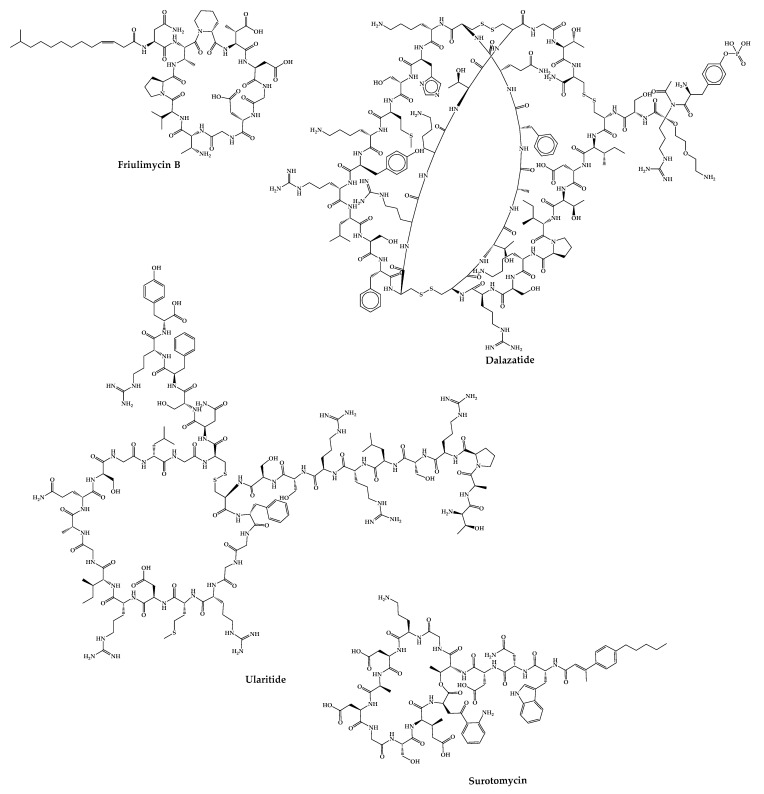
Structures of cyclic peptides in clinical trials whose development has been interrupted.

**Table 1 pharmaceuticals-16-00996-t001:** Cyclic peptides currently in clinical trials.

Peptide Name	Generic Name	Indication	Highest Phase	Company	Source	References
POL6326	Balixafortide	Advanced breast cancers		Spexis	Synthesis	[143]
BL-8040	Motixafortide	HSCs, solid tumors, and AML	III	Bioline RX	Synthesis	[144]
BT1718		Cancer with MT1-MMP expression	I/II	Bicycle TherapeuticsLtd.	Synthesis	[145,146,147]
BT8009		Cancer types, where Nectin-4 is expressed	I/II	Bicycle TherapeuticsLtd.	Synthesis	[148,149]
BT5528		Advanced solid tumors associated with EphA2 expression	I/II	Bicycle TherapeuticsLtd.	Synthesis	[150]
VT1021		Antitumor	II	Vigeo Therapeutics		[151]
ALRN-6924		Chemoprotective agent	Ib	Aileron Therapeutics	Synthesis	[152]
CEND-1		Enhance the efficacy of chemotherapy	II	Cend Therapeutics		[153]
POL7080	Inhaled murepavadin	Antibiotic to treat *Pseudomonas* infections in patients with cystic fibrosis	I	Spexis	Synthesis	[154]
	Thanatin-derivative		Pre-clinical	Spexis/University of Zurich		[155]
CD-101	Rezafungin	Treatment of candidemia and/or invasive candidiasis andinvasive fungal diseases	III	Cidara Therapeutics Inc.	Semi-synthesis	[156]
RA-101495	Zilucoplan	Paroxysmal nocturnal hemoglobinuria, generalized myasthenia gravis	III	UCB Pharma	Synthesis	[157]
AP301	Solnatide	Pulmonary permeability edema	II	Apeptico	Synthesis	[158]
POL6014	Lonodelestat	Cystic fibrosis	II	Santhera Pharmaceuticals	Synthesis	[159]
THR-149		Diabetic macular edema	II	Oxurion NV	Synthesis	[160,161]
PTG-300	Rusfertide	Polycythemia vera	III	Protagonist Therapeutics, Inc.	Synthesis	[162,163]
PN-943		Ulcerative colitis	II	Protagonist Therapeutics, Inc.		[164,165]
PL8177		Ulcerative colitis	II	Palatin TechnologiesInc.	Synthesis	[166]
PA-001		SARSCov-2 infection	I	PeptiAID		[167]
AZP-3813		Acromegaly	IND-Enabling	Amolyt Pharma		[168]
PL9643		Dry eye disease	III	Palatin TechnologiesInc.	Synthesis	[169]
PM90001	Plitidepsin	SARS-CoV-2 infection	III	PharmaMar	Synthesis	[46,170]
APL-2	Pegcetacoplan	ALS, IC-MPGN, and C3G, CAD, and HSCT-TMA		Apellis Pharmaceuticals Inc.		[171]
	Friulimicin B	Antibacterial activity against Gram-positive bacteria	Stopped in phase I in 2008 due to unfavourable pharmacokinetic properties	MerLion Pharmaceuticals GmbH	Nature	[172,173]
CB-183,315	Surotomycin	Antibacterial activity against *Clostridium difficile*	Stopped in phase III in 2019 (no improvement over vancomycin)	Merck		[174]
ShK-186	Dalazatide	Plaque psoriasis	I (finished in May 2015)	Kv1.3	Synthesis	[175]
	Ularitide	Acutely decompensated heart failure	III (finished in October 2018)	Cardiorentis	Synthesis	[176,177]

HSCs—hematopoietic stem cells; AML—acute myeloid leukemia; MT1-MMP—membrane type I matrix metalloproteinase; ALS—amyotrophic lateral sclerosis; IC-MPGN and C3G—immune complex membranoproliferative glomerulonephritis and C3 glomerulopathy; CAD—cold agglutinin disease; HSCT-TMA—hematopoietic stem cell transplant-associated thrombotic microangiopathy.

## Data Availability

Not applicable.

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
