# Peer review of "Cyclic Peptides in Pipeline: What Future for These Great Molecules?"

_pharmaceuticals, 2023, doi:10.3390/ph16070996_

Round 1
Reviewer 1 Report
The authors reviewed the potential of cyclic peptides and general aspects of their synthesis and development are discussed. Furthermore, an overview of already approved cyclic peptides is also given and the cyclic peptides in clinical trials are summarized. The paper meaningful to cyclic drug development, and is useful for the researchers in such field.
The synthesis is summarized in general. The success and failure in peptide development is indeed thought-provoking. I support the publication of this paper.
Some concerns might be addressed, as shown below
1. The failure of some cyclopeptides is due to poor pharmacokinetic parameters. Therefore, I suggest that the authors add a section addressing the pharmacokinetic properties of cyclic peptides, especially for peptides in Phase I/II trials, which could be of great interest.
2. The layout of some structures could be improved. Like the blank spaces.
3. The section of synthesis summary is inadequate. Some synthetic methods are not included in this review, and therefore it is recommended to indicate the limitations of synthetic profiles.
Author Response
Considering: “The authors reviewed the potential of cyclic peptides and general aspects of their synthesis and development are discussed. Furthermore, an overview of already approved cyclic peptides is also given and the cyclic peptides in clinical trials are summarized. The paper meaningful to cyclic drug development, and is useful for the researchers in such field. The synthesis is summarized in general. The success and failure in peptide development is indeed thought-provoking. I support the publication of this paper.”
The authors acknowledge for the valuable revision of the article and the comments made by the reviewer.
Considering: “1. The failure of some cyclopeptides is due to poor pharmacokinetic parameters. Therefore, I suggest that the authors add a section addressing the pharmacokinetic properties of cyclic peptides, especially for peptides in Phase I/II trials, which could be of great interest.”
We understand the reviewer’s point of view, and we improved the discussion about the disadvantageous of peptides when they are used for drug development, mainly concerning pharmacokinetic issues. Also, we highlighted that for many cyclic peptides the poor pharmacokinetic parameters are one of the aim reasons to failure in Phase I/II trials (please see revised manuscript).
Considering: “2. The layout of some structures could be improved. Like the blank spaces.”
We thank the reviewer for this comment, and we improved the layout of some structures.
Considering: “3. The section of synthesis summary is inadequate. Some synthetic methods are not included in this review, and therefore it is recommended to indicate the limitations of synthetic profiles.”
We understand the reviewer’s point of view and we highlighted that we mentioned examples of the most relevant peptide cyclization methods. In addition, the limitations of synthetic profiles were included (please see revised manuscript).
Reviewer 2 Report
The manuscript entitled “Cyclic peptides in pipeline: what future for these great molecules?” by Costa et al. is well written and informative. The reader is introduced to the identification of peptides from different sources, synthesis methods and chemical modification to improve their affinity and stability, phage display screening method. Authors focused to the advantages of cyclic peptides compared to other small molecules and their current applications as therapeutic agents in different medical context.The title matches well with the content and the abstract is concise and clear. The review is well written and clearly presented. References are adequate. The topic matches well with the Special issue in which has been submitted.
I suggest that this review should be accepted for publication in the Pharmaceuticals journal in the present form.
Author Response
Considering: “The manuscript entitled “Cyclic peptides in pipeline: what future for these great molecules?” by Costa et al. is well written and informative. The reader is introduced to the identification of peptides from different sources, synthesis methods and chemical modification to improve their affinity and stability, phage display screening method. Authors focused to the advantages of cyclic peptides compared to other small molecules and their current applications as therapeutic agents in different medical context.
The title matches well with the content and the abstract is concise and clear. The review is well written and clearly presented. References are adequate. The topic matches well with the Special issue in which has been submitted. I suggest that this review should be accepted for publication in the Pharmaceuticals journal in the present form.”
The authors acknowledge for the comments made by the reviewer.